# Multi-state model for predicting ocular progression in acute Stevens-Johnson syndrome/toxic epidermal necrolysis

**Fumie Kinoshita**[1,2], **Isao Yokota**[1,3], **Hiroki Mieno**[4], **Mayumi Ueta**[4], **John Bush**[4], **Shigeru Kinoshita**[4], **Hirohiko Sueki**[5], **Hideo Asada**[6], **Eishin Morita**[7], **Masanori Fukushima**[8], **Chie Sotozono**[4]*, **Satoshi Teramukai**[1], on behalf of The Japanese Research Committee on Severe Cutaneous Adverse Reaction¶

1 Department of Biostatistics, Graduate School of Medical Science, Kyoto Prefectural University of Medicine, Kyoto, Japan, 2 Data Coordinating Center, Department of Advanced Medicine, Nagoya University Hospital, Nagoya, Japan, 3 Department of Biostatistics, Hokkaido University, Sapporo, Japan, 4 Department of Ophthalmology, Kyoto Prefectural University of Medicine, Kyoto, Japan, 5 Department of Dermatology, School of Medicine, Showa University, Tokyo, Japan, 6 Department of Dermatology, Nara Medical University School of Medicine, Kashihara, Japan, 7 Department of Dermatology, Shimane University Faculty of Medicine, Izumo, Japan, 8 Foundation of Learning Health Society Institute, Nagoya, Japan

¶ Membership of the Japanese Research Committee on Severe Cutaneous Adverse Reaction is listed in the Acknowledgments

* csotozon@koto.kpu-m.ac.jp

**Data Availability Statement:** Data cannot be shared publicly because of potentially identifiable information in the dataset. Data are available from the The Japanese Research Committee on Severe

## Abstract

This study aimed to clarify the etiologic factors predicting acute ocular progression in SJS/TEN, and identify patients who require immediate and intensive ophthalmological treatment. We previously conducted two Japanese Surveys of SJS/TEN (i.e., cases arising between 2005–2007 and between 2008–2010), and obtained the medical records, including detailed dermatological and ophthalmological findings, of 230 patients. Acute ocular severity was evaluated as none, mild, severe, and very severe. A multi-state model assuming the Markov process based on the Cox proportional hazards model was used to elucidate the specific factors affecting the acute ocular progression. Our findings revealed that of the total 230 patients, 23 (24%) of 97 cases that were mild at initial presentation worsened to severe/very severe. Acute ocular progression developed within 3 weeks from disease onset. Exposure to nonsteroidal anti-inflammatory drugs (NSAIDs) and younger patient age were found to be statistically significant for the progression of ocular severity from mild to severe/very severe [hazard ratio (HR) 3.83; 95% confidence interval (CI) 1.48 to 9.91] and none to severe/very severe [HR 0.98; 95% CI 0.97 to 0.99], respectively. The acute ocular severity score at worst-condition was found to be significantly correlated with ocular sequelae. Thus, our detailed findings on acute ocular progression revealed that in 24% of SJS/TEN cases with ocular involvement, ocular severity progresses even after initiating intensive treatment, and that in younger-age patients with a history of exposure to NSAIDs, very strict attention must be given to their ophthalmological appearances.

Cutaneous Adverse Reaction (contact via biostat@koto.kpu-m.ac.jpxs) for researchers who meet the criteria for access to confidential data.

**Funding:** This work was supported in part by a Grant-in-Aid for Scientific Research from the Japanese Ministry of Health, Labor and Welfare (https://www.mhlw.go.jp/english, 20FC1035), a research Grant from the Japan Agency for Medical Research and Development(https://www.amed.go.jp/en/, 19ek0109377h, 20ek0109377), and a Research Grant from the Japanese Ministry of Education, Culture, Sports, Science and Technology(https://www.mext.go.jp/en/, 19H03809). The funders had no role in the study design, data collection and analysis, decision to publish, or preparation of the manuscript.

**Competing interests:** The authors have declared that no competing interests exist.

## Introduction

Stevens-Johnson syndrome (SJS) and toxic epidermal necrolysis (TEN) are rare acute inflammatory disorders of the skin and mucous membranes with high mortality rates [1–4]. Reportedly, both SJS and TEN usually develop as a result of adverse reactions to ingested pharmaceutical drugs [5]. A variety of drugs reportedly can lead to the onset of SJS/TEN, with the primary types of those medications being non-steroidal anti-inflammatory drugs (NSAIDs), antibiotics, and anticonvulsants [6–8].

In the acute stage of the disease, more than 50% of SJS/TEN patients present with ocular involvement, and extensive inflammation of the ocular surface is often accompanied by corneal and/or conjunctival epithelial defects. Common signs that appear after the acute stage include persistent epithelial defects, ulceration, and perforation, ultimately developing into corneal, conjunctival, and eyelid cicatricial changes such as neovascularization, opacification, keratinization, and symblepharon [9, 10]. In SJS/TEN cases with severely affected eyes, the disease is accompanied with chronic ocular sequelae, such as visual impairment and severe dry eye (Fig 1) [7, 11, 12]. Thus, it is vital to clinically detect and recognize acute ocular involvement early in order to avoid severe visual dysfunction [7, 10, 13, 14].

In our previous report [7], we retrospectively reviewed Japanese patients who were diagnosed with SJS/TEN between January 2005 and December 2007. In that study, the prognostic factors for severe acute ocular involvement, e.g., ocular-surface epithelial defect and/or pseudomembrane formation, were investigated. Based on our findings, we proposed simple yet specific criteria for recognizing acute ocular involvement in patients with SJS/TEN, and the acute ocular severity score that we developed showed that acute ocular severity was significantly associated with patient age and the specific types of pharmaceutical drugs ingested. In fact, our acute ocular severity score has now been embraced globally, and is now cited in the official SJS/TEN treatment guidelines in both Japan and the UK [15]. In our previous study [7], we used two time-point-specific data references of the acute ocular severity score obtained at 1) the initial day of ophthalmic examination and 2) at the worst day of ocular findings. In that study, only the acute ocular severity scores obtained on the worst day of ocular findings were used for data analysis. Moreover, in our previously reported case-series study involving our successful treatment of Stevens-Johnson syndrome with steroid pulse therapy at disease onset [14], we found that ocular severity became worse within several days after disease onset, even though prompt diagnosis and intensive treatment was administered. It should be noted that to the best of our knowledge, there have been no previously published reports focusing on the progression of ocular severity in acute-phase SJS/TEN patients.

The aim of this present study was to elucidate the specific details of the progression of ocular severity in SJS/TEN patients. To that end, we developed a multi-state model to evaluate the progression of ocular severity in acute-stage SJS/TEN, as multi-state statistical models can be used to simultaneously evaluate the sequence of states (i.e., full analysis of the patient's event history data) [16]. Moreover, we aimed to standardize the identification of SJS/TEN patients who require immediate treatment, as well as the length of time (i.e., the elapsed period) post disease onset in which very strict and careful observation is required. In addition, in order to more deeply clarify the details of our findings, we collected the data of patients who were developed with SJS/TEN from January 2008 to December 2010, and merged that data with our 2005 to 2007 databases.

## Materials and methods

### Patients and data collection

In this retrospective observational study, we reviewed the medical records of Japanese patients who developed SJS/TEN between January 2005 and December 2010. The protocols of this

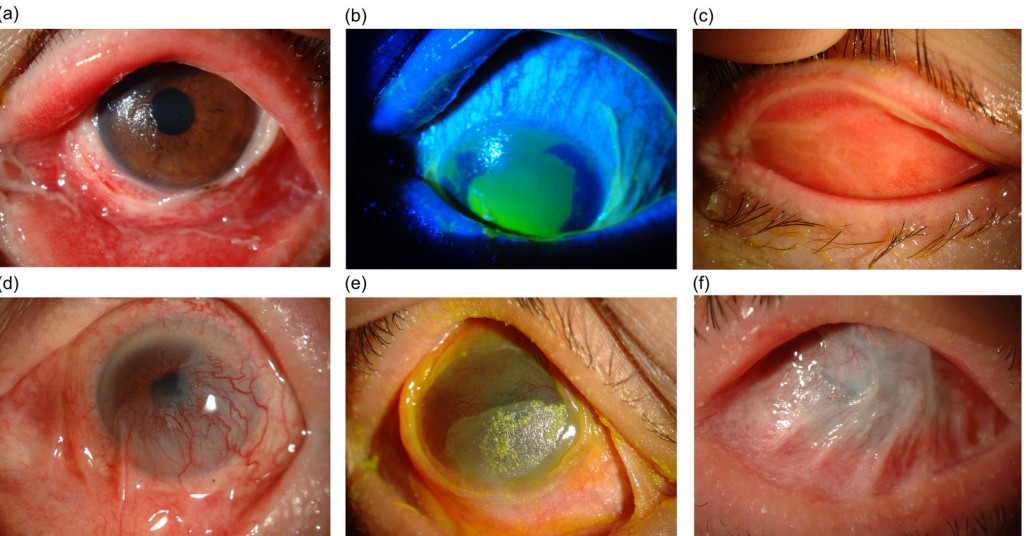

**Fig 1. Ocular surface appearances in Stevens-Johnson syndrome/toxic epidermal necrolysis (SJS/TEN).** At the acute phase, conjunctivitis (**a**), ocular surface epithelial defect (**b**), and/or pseudomembrane formation (**c**) arise in cases with ocular involvement. At the chronic phase, corneal neovascularization and opacification (**d**), keratinization (**e**), and symblepharon (**f**) develop as ocular sequelae.

study were approved by the Ethics Committee and the Institutional Review Board of Kyoto Prefectural University of Medicine, Kyoto, Japan (Approval Nos. RBMR-E-393 and ERB-C-1522), and by the Ethical Review Committee of each affiliated medical institution. Informed consent was obtained from all study participants via the 'opt-out' method, in accordance with the Ethical Guidelines for Medical and Health Research Involving Human Subjects in Japan, and this study was performed in accordance with the tenets set forth in the Declaration of Helsinki.

The analyzed data sets in this study were collected via two independent surveys. The first was a survey that involved Japanese patients who developed SJS/TEN between January 2005 and December 2007. Briefly, a questionnaire was first sent to dermatologists at 607 medical institutions in Japan, of which 212 of those institutions responded. Next, case report forms (CRFs) were sent to the dermatologists who responded. Based on the medical records and the CRFs that were obtained, as well as on the diagnostic criteria developed by the Japanese Ministry of Health, Labour and Welfare in 2005 [7], the patients matching the diagnostic criteria for SJS and TEN was determined. It should be noted that the systemic and dermatologic findings of this study have previously been reported [17]. Next, CRFs aimed at the investigation of ocular involvement were sent to Japanese ophthalmologists who treated SJS/TEN patients upon referral from dermatologists, and those treated patients were selected as the study population. The obtained CRFs of 135 cases (87 SJS patients and 48 TEN patients) were analyzed in order to clarify predictive factors for the acute ocular severity score [7].

The second survey involved Japanese patients who developed SJS/TEN between January 2008 and December 2010. CRFs were sent to ophthalmologists in 51 medical institutions affiliated with the Japanese Ophthalmological Society and/or the Japan Cornea Society, and responses were obtained from 38 medical institutions.

In this present study, the data sets developed from the first Japanese survey (i.e., the CRFs of the patients who were developed as SJS/TEN between 2005 and 2007) were merged with the newly obtained data sets from the second Japanese survey (i.e., the CRFs of the patients who were developed as SJS/TEN between 2008 and 2010), and then analyzed. We excluded patients

in whom the elapsed time from disease onset to the initial-presentation ophthalmological examination or the elapsed time from initial presentation to worst-condition follow-up visit was more than 30 days.

## Acute ocular severity score and systemic severity index score, definition, and measurement

In this study, and in accordance with our previously described methods [7], the ocular surface findings at the acute stage of SJS/TEN were graded as an acute ocular severity score ranging from Grade 0 to 3, in which the grades were termed "none", "mild", "severe", and "very severe", respectively (Fig 1). Conjunctival hyperemia, which indicates ocular surface inflammation, was assessed as Grade 1. Eyes with accompanying pseudomembrane formation or an ocular surface epithelial defect were assessed as Grade 2. Eyes with both pseudomembrane formation and an ocular surface epithelial defect were assessed as Grade 3. The acute ocular severity scores at the day of worst severity during the acute stage were then documented.

The systemic severity index score in this study was a summed score (see S1 Table), with the maximum score being 14 [18, 19]. Moreover, the systemic severity index subscore was defined as a summed score that includes the score for surface area of skin lesions on the body, except for the score for ocular lesion, with the maximum score being 11.

## Ocular sequelae definition

The obtained CRFs included data on whether or not there was any occurrence of ocular sequelae at the chronic stage. Visual impairment was defined as patients with a visual impairment of less than 20/20, and dry eye-related data was also obtained.

## Elapsed time from disease onset to initial presentation or the follow-up visit in which the worst condition was observed

The date of disease onset was defined as the date in which erythema initially developed, and if that date was unknown, then the date of the onset of high fever was used. The date of initial presentation was defined as the number of elapsed days from disease onset until the first examination by an ophthalmologist (i.e., the initial presentation).

We selected patients who had an acute ocular severity score of either "none" or "mild" at initial presentation and "mild" or "severe/very severe" at the worst-condition follow-up visit, and then calculated the elapsed time (i.e., in days) from the date of initial presentation to the date of worst-condition follow-up visit.

## Statistical analysis

To summarize the data, the median and range for continuous variables and frequencies (%) for categorical variables were used. We compared the acute ocular severity score of Grades 2 or 3 to Grades 0 or 1 using the Wilcoxon rank sum test for the continuous variables and Fisher's exact test for the categorical variables.

The specific states and transitions for the acute ocular severity were examined using a multi-state model and are shown in Fig 2 [16, 20–23]. In this study, the following three specific states were included: 1) State 0: Initial state before disease onset or no ocular involvement post onset, 2) State 1: Acute ocular severity being mild, and 3) State 2: Acute ocular severity being severe or very severe. The definition of "mild", "severe", and "very severe" at each visit were the same as used for the acute ocular severity score. To estimate the adjusted hazard ratios (HRs) on each transition assuming the Markov process, a multivariate Cox proportional

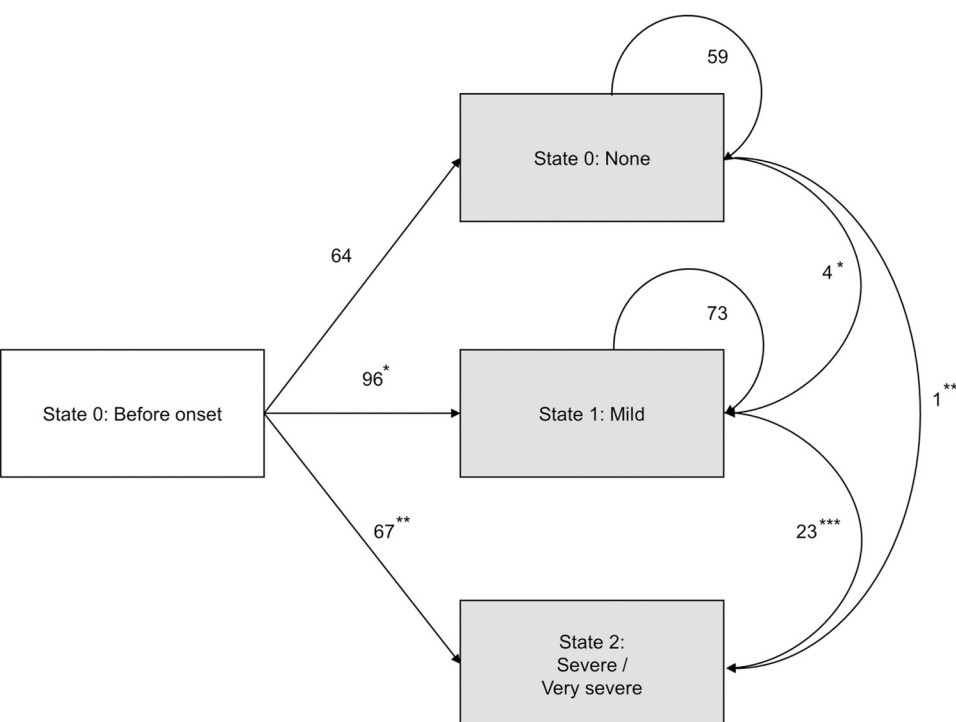

**Fig 2. Diagram showing the multi-state model of acute ocular severity progression in the SJS/TEN patients in this study.** The arrows indicate the changes among the three states. The numbers next to each arrow indicate the number of cases. The white rectangular box on the left-hand side shows the initial state prior to disease onset. Asterisks indicate the following events of transition shown in Table 6; * = transition from none to mild, ** = transition from none to severe / very severe, and *** = transition from mild to severe / very severe.

hazards model was used. The elapsed time to initial presentation or worst-condition follow-up visit, in which the patients were evaluated as acute ocular severity, was used. Patients who did not progress after each state were censored at the date of the final follow-up examination. Moreover, the transition probabilities of each state at each time-point were obtained from the Aalen-Johansen transition estimates [24]. Transition probabilities for patients stratified by age and history of exposure to NSAIDs were also estimated.

All statistical tests were two-sided, and a *P*-value of <0.05 was considered statistically significant. All analyses were performed using SAS version 9.4 (SAS Institute, Inc., Cary, NC, USA) statistics software.

## Results

### Patient demographics

In the first survey, which involved Japanese patients who were developed with SJS/TEN between January 2005 and December 2007, data was collected from 150 patients. Eight ineligible patients and 7 patients with incomplete data were excluded from the analysis. In the second survey, which involved Japanese patients who were developed with SJS/TEN between January 2008 and December 2010, data was collected from 115 patients. One ineligible patient and 2 patients with incomplete data were excluded from the analysis. In a total of 247 patients (135 patients in the first survey and 112 patients in the second survey) seen at 79 medical institutions, the distribution of the elapsed time from the date of disease onset to the date of initial examination by an ophthalmologist is shown in Fig 3, and the median time (range) was 4 days

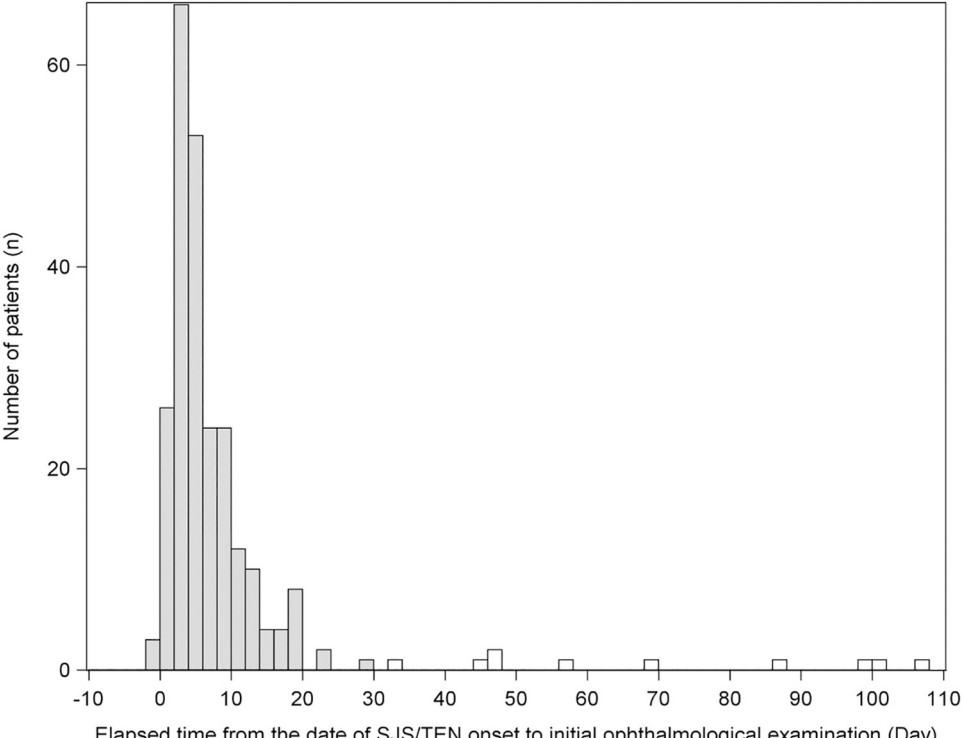

**Fig 3. Elapsed time from the date of SJS/TEN onset to initial ophthalmological examination.** The grey bars indicate the patients who underwent ophthalmological examination within 30 days post disease onset.

(-2 to 107 days). It should be noted that the patients with "negative" days were those who underwent an ophthalmological examination prior to the development of erythema.

Of the total 247 patients, 17 patients in whom the period of elapsed time from disease onset to initial ophthalmological examination or the period of worst-condition follow-up visit was long were excluded. In 230 patients [96 males (42%) and 134 females (58%)], the median age was 57.0 years (range: 5 to 90 years), and 161 were SJS patients (70%) and 69 were TEN patients (30%). Of the 230 patients, 213 (93%) underwent ophthalmological examination within 2 weeks post disease onset.

## The progression of acute ocular severity

The change of acute ocular severity from initial presentation to worst-condition follow-up visit is shown in Table 1. Of the total 230 patients, 64 were scored as "none" at the initial presentation, and 92% (59/64) remained scored as "none" at the final follow-up examination. In contrast, 97 patients were scored as "mild" at the initial presentation, and in 24% (23/97) of these patients, the score of their condition worsened to "severe" or "very severe". Among those patients, the median elapsed time period from initial presentation to worst-condition follow-up visit was 4 days (range: 1 to 18 days).

## Association between patient characteristics and acute ocular severity

In subsequent analyses, Grades 2 and 3 were handled as one category (i.e., "Grade 2/3"), as those two grades imply a severe amount of ocular surface inflammation due to those grades being closely related to visual dysfunction at the chronic phase [7]. The patient characteristics

**Table 1. The change of acute ocular severity.** Acute ocular severity at initial presentation was graded as acute ocular severity at worst-condition follow-up visit.

| Acute ocular severity at initial visit | Acute ocular severity at worst-condition follow-up visit N (%) | | | | Total N |
| --- | --- | --- | --- | --- | --- |
| | **none** | **mild** | **severe** | **very severe** | |
| none | 59 (92.2) | 4 (6.3) | 1 (1.6) | 0 (0.0) | 64 |
| mild | 0 (0.0) | 74 (76.3) | 19 (19.6) | 4 (4.1) | 97 |
| severe | 0 (0.0) | 0 (0.0) | 43 (86.0) | 7 (14.0) | 50 |
| very severe | 0 (0.0) | 0 (0.0) | 0 (0.0) | 19 (100.0) | 19 |
| Total | 59 | 78 | 63 | 30 | 230 |

Percentage was calculated as the number of patients at each category / number of all patients in each acute ocular severity at initial presentation.

at diagnosis by acute ocular severity score at worst-condition follow-up visit are shown in Table 2. The acute ocular severity score was Grade 0 in 59 patients (26%), Grade 1 in 78 patients (34%), and Grade 2/3 in 93 patients (40%). The median patient age in Grades 0, 1, and

**Table 2. Patient characteristics.**

| | | Acute Ocular Severity Score Median (range) or N (%) | | | P-value |
| --- | --- | --- | --- | --- | --- |
| | | **Grade 0 (none)** | **Grade 1 (mild)** | **Grade 2/3 (severe/very severe)** | |
| | | **N = 59** | **N = 78** | **N = 93** | |
| Age at onset | | 62.0 (18–88) | 58.5 (12–90) | 50.0 (5–81) | 0.003 |
| Sex | male / female | 20 (33.9) | 41 (52.6) | 35 (37.6) | 0.341 |
| Diagnosis | TEN / SJS | 15 (25.4) | 22 (28.2) | 32 (34.4) | 0.244 |
| Causative drug | | | | | |
| NSAIDs | | 15 (25.4) | 16 (20.5) | 39 (41.9) | 0.002 |
| Cold remedies | | 3 (5.1) | 10 (12.8) | 18 (19.4) | 0.048 |
| Antibiotics | | 15 (25.4) | 21 (26.9) | 19 (20.4) | 0.347 |
| Anticonvulsants | | 13 (22.0) | 22 (28.2) | 19 (20.4) | 0.429 |
| Treatment for gout | | 2 (3.4) | 11 (14.1) | 12 (12.9) | 0.518 |
| Elapsed time period (in days) from disease onset to first-visit ophthalmological examination | | 5.0 (0–23) | 5.0 (-2–29) | 4.0 (-1–19) | 0.017 |
| Systemic severity index subscore | | 5 (1–9) | 5 (1–11) | 6 (1–11) | 0.008 |
| Systemic severity | | | | | |
| Fever | ≥38.0 | 51 (86.4) | 61 (78.2) | 76 (81.7) | 1.000 |
| Ratio of body surface area of skin lesions | <10% | 44 (74.6) | 56 (71.8) | 61 (65.6) | 0.029 |
| | 10%≤ <30% | 5 (8.5) | 9 (11.5) | 4 (4.3) | |
| | 30%≤ | 10 (17.0) | 13 (16.7) | 28 (30.1) | |
| Epidermal detachment | | 24 (40.7) | 26 (33.3) | 38 (40.9) | 0.581 |
| Labial and/or oral lesions | Oral or labial erosive lesions alone | 17 (28.8) | 18 (23.1) | 19 (20.4) | 0.008 |
| | Labial erosive lesions with bloody scales alone | 22 (37.3) | 17 (21.8) | 27 (29.0) | |
| | Oral diffuse erosive lesions with bloody scales | 11 (18.6) | 26 (33.3) | 41 (44.1) | |
| Genital involvement | | 24 (40.7) | 33 (42.3) | 45 (48.4) | 0.345 |
| Respiratory dysfunction | | 2 (3.4) | 13 (16.7) | 13 (14.0) | 0.540 |
| Liver dysfunction | | 16 (27.1) | 15 (19.2) | 27 (29.0) | 0.283 |

Proportion was calculated as the number of patients at each category / number of all patients in each acute ocular severity score. TEN: toxic epidermal necrolysis; SJS: Stevens-Johnson syndrome; NSAIDs: nonsteroidal anti-inflammatory drugs

2/3 was 62.0 years (range: 18 to 88 years), 58.5 years (range: 12 to 90 years), and 50.0 years (range: 5 to 81 years), respectively. NSAIDs were the ingested drugs in 25% of the Grade 0 patients and in 21% of the Grade 1 patients, yet were the ingested drugs in 42% of the Grade 2/3 patients. Cold remedies were the ingested drugs in 5%, 13%, and 19% of the Grade 0, Grade 1, and Grade 2/3 patients, respectively. The ratio of body surface area of skin lesions and the severity of oral mucosal involvement was found to be associated with the acute ocular severity score.

The median follow-up time from initial examination by an ophthalmologist in Grades 0, 1, and 2/3 was 5 days, 21.5 days, and 128 days, respectively.

### Correlation between acute ocular severity scores and outcomes

Ten patients were excluded from the analysis of ocular sequelae due to the related data either being missing (n = 4 cases) or patient death (n = 6 cases). The outcomes and ocular sequelae by acute ocular severity score at worst-condition follow-up visit are shown in Table 3. The incidence of patient death was found to not be associated with the acute ocular severity score. In contrast, the ocular sequelae were found to be associated with the acute ocular severity score by symptom. For example, while visual impairment was found to have occurred in only 2% of the Grade 0 patients, its occurrence in the Grade 1 and Grade 2/3 patients was 3% and 20%, respectively. Similarly, while dry eye was found to have occurred in only 5% of the Grade 0 patients, its occurrence in the Grade 1 and Grade 2/3 patients was 19% and 48%, respectively.

On the other hand, 7 patients in whom analysis was excluded due to the initial ophthalmological examination being performed within 30 days yet the elapsed time between initial presentation and worst-condition follow-up visit was more than 30 days, were graded as severe on the acute ocular severity score, and ocular sequelae was present in all patients (Table 4). In those patients, the mean elapsed time period from disease onset to initial ophthalmological examination was 7 days (range: 1–16 days). Moreover, in 10 patients in whom analysis was excluded due to the elapsed time from disease onset to initial ophthalmological examination being more than 30 days, ocular sequelae occurred in all but 1 patient with a grade of 0 (Table 5).

### Prognostic factors of the progression of ocular severity and transition probabilities via multi-state model analysis

For multi-state model analysis of the prognostic factors of the progression of ocular severity and transition probabilities, of the above-described 230 patients, 227 patients were included,

**Table 3. Correlation between outcomes or ocular sequelae and acute ocular severity score at worst-condition follow-up visit.**

| Outcome | | Acute Ocular Severity Score N (%) | | | P-value |
|---|---|---|---|---|---|
| | | Grade 0 (none) | Grade 1 (mild) | Grade 2/3 (severe/very severe) | |
| | | N = 59 | N = 78 | N = 93 | |
| Death | | 2 (3.4) | 6 (7.7) | 7 (7.5) | 0.600 |
| Ocular sequelae | | 3 (5.1) | 14 (18.7) | 46 (53.5) | <0.001 |
| Visual disturbance | 20/20–20/200 of BCVA | 1 (1.7) | 1 (1.3) | 16 (18.6) | <0.001 |
| | Worse than 20/200 of BCVA | 0 (0.0) | 1 (1.3) | 1 (1.2) | |
| Severity of dry eye | Mild | 3 (5.1) | 11 (14.7) | 28 (32.6) | <0.001 |
| | Moderate | 0 (0.0) | 2 (2.7) | 9 (10.5) | |
| | Severe | 0 (0.0) | 1 (1.3) | 4 (4.7) | |
| Missing data about ocular sequelae | | 0 | 3 | 7 | |

Proportion was calculated as number of patients at each category / as number of all patients in each acute ocular severity score. Eleven patients were missing about ocular sequelae. P-value: Fisher's exact test

**Table 4. Outcomes and ocular sequelae of the patients in whom the time from disease onset to the initial ophthalmological examination was within 30 days but the period of elapsed time from initial presentation to worst-condition follow-up visit was more than 30 days.**

| Outcome | | Acute Ocular Severity Score N (%) | | |
|---|---|---|---|---|
| | | Grade 0 (none) | Grade 1 (mild) | Grade 2/3 (severe/very severe) |
| | | N = 0 | N = 0 | N = 7 |
| Death | | 0 (0.0) | 0 (0.0) | 0 (0.0) |
| Ocular sequelae | | 0 (0.0) | 0 (0.0) | 7 (100.0) |
| Visual disturbance | 20/20–20/200 of BCVA | 0 (0.0) | 0 (0.0) | 2 (28.6) |
| | Worse than 20/200 | 0 (0.0) | 0 (0.0) | 4 (57.1) |
| Severity of dry eye | Mild | 0 (0.0) | 0 (0.0) | 1 (14.3) |
| | Moderate | 0 (0.0) | 0 (0.0) | 1 (14.3) |
| | Severe | 0 (0.0) | 0 (0.0) | 4 (57.1) |
| Missing data about ocular sequelae | | 0 | 0 | 0 |

while 3 patients were excluded from analysis due to being evaluated as acute ocular involvement prior to the onset of erythema.

The changes between severity states (i.e., from one to another state) are indicated by the arrows shown in Fig 2, with the number next to each arrow indicating the number of patients. Analysis of the transition from the state of "none" to "mild" revealed that females (HR 0.65; 95% CI 0.43 to 0.97) had a lower probability of transition compared to males (Table 6). Moreover, analysis of the transition from the state of "none" to "severe/very severe" revealed that patients of a younger age at disease onset (HR 0.98; 95% CI 0.97 to 0.99) had a higher probability of transition. However, exposure to NSAIDs was found to have a highly negative effect on the transition from "mild" to "severe/very severe" (HR 3.83; 95% CI 1.48 to 9.91).

Stacked transition probabilities over a 30-day period post disease onset for patients in different age categories (i.e., patients under 45 years old vs. patients aged 45 years and over), as well as for patients with or without exposure to NSAIDs, are shown in Fig 4. In the patients who were under 45 years old and who had a history of exposure to NSAIDs, there was high probability rate of progressing to the state of "severe/very severe". Moreover, our findings revealed that the patients who had a history of exposure to NSAIDs had a high probability of experiencing a worsening of the ocular severity after the "mild" state, and that the progression mainly occurred within 3 weeks post disease onset.

**Table 5. Outcomes and ocular sequelae of the patients in whom the period of elapsed time from disease onset to the initial ophthalmological examination was more than 30 days.**

| Outcome | | Acute Ocular Severity Score N (%) | | |
|---|---|---|---|---|
| | | Grade 0 (none) | Grade 1 (mild) | Grade 2/3 (severe/very severe) |
| | | N = 1 | N = 5 | N = 4 |
| Death | | 0 (0.0) | 1 (25.0) | 0 (0.0) |
| Ocular sequelae | | 0 (0.0) | 4 (100.0) | 4 (100.0) |
| Visual disturbance | 20/20–20/200 of BCVA | 0 (0.0) | 2 (50.0) | 2 (50.0) |
| | Worse than 20/200 | 0 (0.0) | 1 (25.0) | 2 (50.0) |
| Severity of dry eye | Mild | 0 (0.0) | 1 (25.0) | 1 (25.0) |
| | Moderate | 0 (0.0) | 2 (50.0) | 0 (0.0) |
| | Severe | 0 (0.0) | 1 (25.0) | 2 (50.0) |
| Missing data about ocular sequelae | | 0 | 1 | 0 |

Proportion was calculated as number of patients at each category / as number of all patients in each acute ocular severity score.

**Table 6. HRs and 95% CIs for prognostic factors of each transition by multi-state model analysis.**

|  | Category | Transition from none to mild | | | Transition from none to severe / very severe | | | Transition from mild to severe / very severe | | |
| --- | --- | --- | --- | --- | --- | --- | --- | --- | --- | --- |
|  |  | N = 227 (Number of events: 100) | | | N = 227 (Number of events: 68) | | | N = 100 (Number of events: 23) | | |
|  |  | HR | 95% CI | *P*-value | HR | 95% CI | *P*-value | HR | 95% CI | *P*-value |
| Age at onset |  | 0.99 | 0.98–1.00 | 0.123 | 0.98 | 0.97–0.99 | 0.004 | 0.99 | 0.97–1.02 | 0.573 |
| Sex | female (vs. male) | 0.65 | 0.43–0.97 | 0.033 | 1.34 | 0.79–2.26 | 0.275 | 1.30 | 0.52–3.22 | 0.574 |
| Diagnosis | TEN (vs. SJS) | 1.03 | 0.55–1.92 | 0.931 | 0.51 | 0.24–1.12 | 0.094 | 1.18 | 0.34–4.07 | 0.789 |
| Causative drug |  |  |  |  |  |  |  |  |  |  |
|  NSAIDs | yes (vs. no) | 1.12 | 0.68–1.83 | 0.655 | 1.36 | 0.79–2.32 | 0.265 | 3.83 | 1.48–9.91 | 0.006 |
|  Cold remedies | yes (vs. no) | 1.14 | 0.61–2.13 | 0.683 | 1.29 | 0.67–2.50 | 0.444 | 0.59 | 0.18–1.92 | 0.384 |
|  Antibiotics | yes (vs. no) | 1.41 | 0.89–2.24 | 0.139 | 0.62 | 0.33–1.19 | 0.149 | 0.81 | 0.29–2.29 | 0.689 |
|  Anticonvulsants | yes (vs. no) | 1.37 | 0.84–2.23 | 0.204 | 0.73 | 0.38–1.42 | 0.352 | 0.94 | 0.32–2.78 | 0.917 |
|  Treatment for gout | yes (vs. no) | 1.80 | 0.97–3.34 | 0.063 | 1.62 | 0.70–3.73 | 0.257 | 1.79 | 0.39–8.34 | 0.457 |
| Systemic severity index subscore |  | 0.95 | 0.84–1.07 | 0.358 | 1.16 | 0.99–1.36 | 0.072 | 1.19 | 0.92–1.52 | 0.180 |

HR: hazard ratio; CI: confidence interval; TEN: toxic epidermal necrolysis; SJS: Stevens-Johnson syndrome; NSAIDs: nonsteroidal anti-inflammatory drugs

## Discussion

To the best of our knowledge, the findings in this study are the first to reveal the details of ocular progression during the acute stage of SJS/TEN via longitudinal data. In this study, multi-

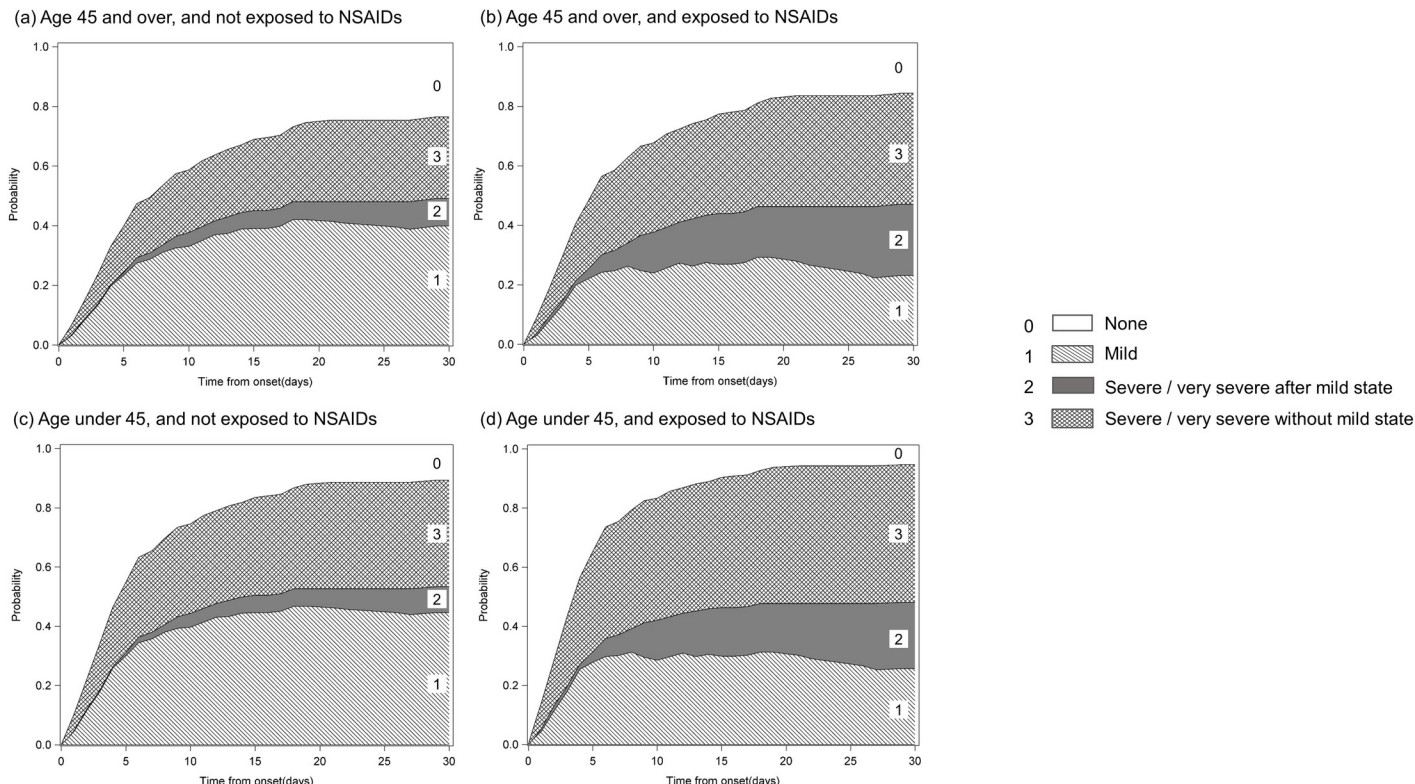

**Fig 4. Stacked prediction probabilities of acute ocular severity in SJS/TEN patients stratified by age categories and with or without exposure to nonsteroidal anti-inflammatory drugs (NSAIDs).** (**a**) Age 45 and over with no exposure to NSAIDs. (**b**) Age 45 and over with exposure to NSAIDs. (**c**) Age under 45 with no exposure to NSAIDs. (**d**) Age under 45 with exposure to NSAIDs.

state model analysis clearly showed the prognostic factors of the progression of acute ocular severity in SJS/TEN patients with ocular involvement, thus certifying that it is a model that can be used effectively for the analysis of event history, i.e., data obtained by observing individuals over time. Although this model has previously been applied in the other fields of research [25–28], to the best of our knowledge, this is the first report in which a multi-state model was applied for the analysis of the ocular progression in SJS/TEN patients. In this present study, we theorized that the multi-state model analysis would allow us to successfully identify the covariates that have an effect on the progression of acute ocular severity in SJS/TEN patients post disease onset, as well as to predict the probabilities of the progression.

Interestingly, our analysis of acute ocular severity revealed that patient age at disease onset has an effect on the involvement of the state of "severe/very severe". Exposure to NSAIDs had a strong influence on the progression of acute ocular severity after the "mild" state, thus illustrating that the patients with a history of exposure to NSAIDs had a high probability of progressing to an acute ocular severity, even if the severity at the time of initial presentation was mild.

In addition, we estimated the predicted probability of the progression of ocular severity by different age groups, and whether or not the patient had a history of exposure to NSAIDs. The patient 'cut-off' age was set to 45 years based on clinical impression. The probability of progression according to the patient's age and history of exposure to NSAIDs are clearly shown in Fig 4. Interestingly, the progression of ocular severity mainly occurred within 3 weeks from disease onset in all groups. Based on these findings, strict and careful examination of ocular involvement within 3 weeks post disease onset is vital, especially in SJS/TEN patients who are young or who have a history of exposure to NSAIDs. The findings in this current study strongly support those presented in a previous report, which indicated that early treatment for ocular involvement was key to obtaining an acceptable visual prognosis and that the critical period for ophthalmological care might be within 7 days post disease onset [29].

In this study, the majority of the patients (i.e., 92%) who had an ocular severity score of "none" at initial presentation experienced no ocular involvement throughout the acute phase. In contrast, some patients (i.e., 24%) experienced a worsening of ocular severity after the "mild" state at initial presentation. Thus, SJS/TEN patients with any ocular involvements should be strictly followed in the acute phase.

It should also be noted that the findings in this present study are the first to illustrate in a relatively large sample size that the severity of oral mucosal involvement is associated with the acute ocular severity. In contrast, however, we found that a respiratory disorder is not related to the severity of acute ocular involvement.

In this present study, 104 of the total 247 SJS/TEN patients seen at 79 medical institutions had "severe/very severe" ocular involvement at the acute phase. We previously reported that common cold medicine (including NSAIDs) related to SJS/TEN with severe ocular involvements was strongly associated with HLA-A*02:06 and significantly associated with HLA-B*44:03 in Japanese patients, and that the HLA allele was insignificant in the SJS/TEN cases without any severe ocular involvement [30]. Thus, a genetic predisposition, such as an HLA allele, might be different in SJS/TEN cases with- and without severe ocular involvements. Moreover, it is known that certain causative drugs may render different genetic predispositions, such as specific HLA alleles.

It should be noted that in this Japanese study, the elapsed period of time between disease onset and the initial ophthalmological examination might have been somewhat short in comparison to that in the previously published reports from other countries. However, the findings in this present study did reveal that the proportion of ocular sequelae was high in the patients in whom the time from disease onset to the initial ophthalmological examination was over 30 days. The official Japanese guidelines for SJS/TEN cases published in 2005 state that

ophthalmologists should be included in the treatment team, and the findings in this present study strongly support the importance of early ophthalmological examination post disease onset.

It should also be noted that this study did have some limitations. First, this was a retrospective study, and second, the number of cases was relatively small. Thus, we had a limited amount of data to examine the relationship between the systemic and ophthalmic treatment and visual prognosis.

In conclusion, via the use of a multi-state model, we showed that acute-phase SJS/TEN patients with a history of exposure to NSAIDs have a high probability of experiencing a worsening of acute ocular severity after the "mild" state, and that progression to the "severe/very severe" state in young patients tends to be rapid, i.e., usually within 3 weeks post disease onset. Thus, SJS/TEN patients should be examined by ophthalmologists from the day of disease onset, and should be closely followed for at least 3 weeks thereafter, especially those who present with ocular involvement at initial presentation, even if it is mild, and who have a history of exposure to NSAIDs.

## Supporting information

**S1 Table. Systemic severity index score for SJS/TEN.**
(DOCX)

## Acknowledgments

We wish to thank the Translational Research Center for Medical Innovation, Foundation for Biomedical Research and Innovation, Kobe, Japan for project and data management. We also wish to thank you to the members of The Japanese Research Committee on Severe Cutaneous Adverse Reaction: Riichiro Abe, Hideo Hashizume, Michiko Kurosawa, Fumi Miyagawa, Taisei Mushiroda. Hiroyuki Niihara, Takashi Nomura, Manabu Ohyama, Hayato Takahashi, Mikiko Tohyama, Hideaki Watanabe, Yukie Yamaguchi, Lead author of Consortium: Hideo Asada asadah@naramed-u.ac.jp

## Author Contributions

**Conceptualization:** Fumie Kinoshita, Mayumi Ueta, Shigeru Kinoshita, Hirohiko Sueki, Hideo Asada, Eishin Morita, Chie Sotozono.

**Data curation:** Fumie Kinoshita, Chie Sotozono, Satoshi Teramukai.

**Formal analysis:** Fumie Kinoshita, Isao Yokota, Satoshi Teramukai.

**Funding acquisition:** Chie Sotozono.

**Investigation:** Hiroki Mieno, Mayumi Ueta, Shigeru Kinoshita, Hirohiko Sueki, Hideo Asada, Eishin Morita, Chie Sotozono.

**Project administration:** Chie Sotozono.

**Supervision:** Chie Sotozono.

**Visualization:** Fumie Kinoshita.

**Writing – original draft:** Fumie Kinoshita.

**Writing – review & editing:** Isao Yokota, Hiroki Mieno, Mayumi Ueta, John Bush, Shigeru Kinoshita, Hirohiko Sueki, Hideo Asada, Eishin Morita, Masanori Fukushima, Chie Sotozono, Satoshi Teramukai.

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
