## [Decision Letter · Decision Letter 0]

16 Nov 2021

Multi-state model for predicting ocular progression in acute Stevens-Johnson syndrome/toxic epidermal necrolysis

PONE-D-21-26387

Dear Dr. Sotozono,

We’re pleased to inform you that your manuscript has been judged scientifically suitable for publication and will be formally accepted for publication once it meets all outstanding technical requirements.

Kind regards,

Abdelrahman M. Elhusseiny

Academic Editor

PLOS ONE

Journal Requirements:

1. PLOS requires an ORCID iD for the corresponding author in Editorial Manager on papers submitted after December 6th, 2016. Please ensure that you have an ORCID iD and that it is validated in Editorial Manager. To do this, go to ‘Update my Information’ (in the upper left-hand corner of the main menu), and click on the Fetch/Validate link next to the ORCID field. This will take you to the ORCID site and allow you to create a new iD or authenticate a pre-existing iD in Editorial Manager. Please see the following video for instructions on linking an ORCID iD to your Editorial Manager account: https://www.youtube.com/watch?v=_xcclfuvtxQ

2. One of the noted authors is a group or consortium [The Japanese Research Committee on Severe Cutaneous Adverse Reaction]. In addition to naming the author group, please list the individual authors and affiliations within this group in the acknowledgments section of your manuscript. Please also indicate clearly a lead author for this group along with a contact email address.’ 

3. We note that Figure 1 in your submission contain copyrighted images. All PLOS content is published under the Creative Commons Attribution License (CC BY 4.0), which means that the manuscript, images, and Supporting Information files will be freely available online, and any third party is permitted to access, download, copy, distribute, and use these materials in any way, even commercially, with proper attribution. For more information, see our copyright guidelines: http://journals.plos.org/plosone/s/licenses-and-copyright.

Reviewers' comments:

Reviewer's Responses to Questions

**Comments to the Author**

1. Is the manuscript technically sound, and do the data support the conclusions?

Reviewer #1: Yes

Reviewer #2: Yes

2. Has the statistical analysis been performed appropriately and rigorously? 

Reviewer #1: Yes

Reviewer #2: Yes

3. Have the authors made all data underlying the findings in their manuscript fully available?

Reviewer #1: No

Reviewer #2: Yes

4. Is the manuscript presented in an intelligible fashion and written in standard English?

Reviewer #1: Yes

Reviewer #2: Yes

5. Review Comments to the Author

Reviewer #1: In addition to the previously published report (reference #7), this is a potentially useful statistical model to predict SJS/TEN outcomes and reveal the chronological details of ocular progression during the acute stage of the disease. The study is well designed, analyzed and written. Congratulations on the study!

Reviewer #2: Your findings on acute ocular progression revealed that in 24% of SJS/TEN cases with ocular

involvement, ocular severity progresses even after initiating intensive treatment, and that in younger age patients with a history of exposure to NSAIDs, is interesting.

6. PLOS authors have the option to publish the peer review history of their article (what does this mean?). If published, this will include your full peer review and any attached files.

Reviewer #1: **Yes: **Taher Eleiwa

Reviewer #2: No

---

## [Editor Report · Acceptance letter]

15 Dec 2021

PONE-D-21-26387 

Multi-state model for predicting ocular progression in acute Stevens-Johnson syndrome/toxic epidermal necrolysis 

Dear Dr. Sotozono:

I'm pleased to inform you that your manuscript has been deemed suitable for publication in PLOS ONE. Congratulations! Your manuscript is now with our production department. 

Kind regards, 

on behalf of

Dr. Abdelrahman M. Elhusseiny 

Academic Editor

PLOS ONE